# Data augmentation in Bayesian neural networks and the cold posterior effect

**Seth Nabarro**[*1]     **Stoil Ganev**[*2]     **Adrià Garriga-Alonso**[3]

**Vincent Fortuin**[3,4]     **Mark van der Wilk**[†1]     **Laurence Aitchison**[†2]

[1]Department of Computing, Imperial College London
[2]Department of Computer Science, University of Bristol
[3]Department of Engineering, University of Cambridge
[4]Department of Computer Science, ETH Zürich

## Abstract

Bayesian neural networks that incorporate data augmentation implicitly use a "randomly perturbed log-likelihood [which] does not have a clean interpretation as a valid likelihood function" (Izmailov et al. 2021). Here, we provide several approaches to developing principled Bayesian neural networks incorporating data augmentation. We introduce a "finite orbit" setting which allows valid likelihoods to be computed exactly, and for the more usual "full orbit" setting we derive multi-sample bounds tighter than those used previously. These models cast light on the origin of the cold posterior effect. In particular, we find that the cold posterior effect persists even in these principled models incorporating data augmentation. This suggests that the cold posterior effect cannot be dismissed as an artifact of data augmentation using incorrect likelihoods.

## 1 INTRODUCTION

The cold posterior effect [CPE; Wenzel et al., 2020] is the surprising observation that performance in neural networks is not optimal when we use the usual Bayesian posterior [Kolmogorov, 1950, Savage, 1954, Jaynes, 2003],

$$\mathrm{P}\left(\mathbf{w}|\mathbf{y}, \mathbf{X}\right) \propto \mathrm{P}\left(\mathbf{w}\right) \mathrm{P}\left(\mathbf{y}|\mathbf{w}, \mathbf{X}\right) \qquad (1)$$

where $\mathbf{w}$ are the neural network weights, $\mathbf{X}$ is all inputs (typically images), and $\mathbf{y}$ is all outputs (typically class labels). Instead, we get better performance when using a "cold" posterior, i.e. the posterior taken to the power of $1/T$ where $T < 1$,

$$\mathrm{Q}\left(\mathbf{w}\right) \propto \left(\mathrm{P}\left(\mathbf{w}\right) \mathrm{P}\left(\mathbf{y}|\mathbf{w}, \mathbf{X}\right)\right)^{1/T}. \qquad (2)$$

The origin of the CPE is by now highly contentious, with three leading potential explanations [Noci et al., 2021]. The first hypothesis is that the process of data curation for popular datasets such as CIFAR-10 and ImageNet [Krizhevsky et al., 2009, Deng et al., 2009] involves multiple annotators agreeing upon the label for each image. In that case, there are in effect multiple labels for each image, which inflates the likelihood (but not the prior) term causing a "cooler" posterior [Adlam et al., 2020, Aitchison, 2020]. Second, the prior is always misspecified, and prior misspecification is known to induce cold posterior-like effects in specific (non-neural network) models [Grünwald, 2012, Grünwald et al., 2017, Adlam et al., 2020], which might give an explanation for the CPE in neural networks [Wenzel et al., 2020, Fortuin et al., 2021b]. However, Fortuin et al. [2021b] showed that better priors do not always reduce the size of the CPE, but can actually increase it. In particular, they found that incorporating spatial correlations in convolutional filters improved the performance of a ResNet trained on CIFAR-10, but also increased the magnitude of the CPE. The third possible explanation is that the CPE is an artifact of data augmentation [DA; Wenzel et al., 2020, Izmailov et al., 2021], as DA gives a "randomly perturbed log-likelihood [which] does not have a clean interpretation as a valid likelihood function" [Izmailov et al., 2021]. This is supported by observations in which the CPE only exists with DA, and disappears without DA [Wenzel et al., 2020, Fortuin et al., 2021b, Izmailov et al., 2021]. Of course it is quite possible that practical CPEs arise from a complex combination of these causes [Aitchison, 2020, Noci et al., 2021].

In spite of this controversy, recent work on the CPE agrees that it is important to investigate integrating DA with Bayesian neural networks (BNNs), and to examine the interaction with the CPE. From Noci et al. [2021]: "It remains an interesting open problem how to properly account for data augmentation in a Bayesian sense." And from Izmailov et al. [2021]: "Data augmentation cannot be naively incorporated in the Bayesian neural network model." and "We leave incorporating data augmentation ... as an exciting direction of

---

* equal contribution
† equal contribution

*Accepted for the 38ᵗʰ Conference on Uncertainty in Artificial Intelligence* (UAI 2022).

future work."

Perhaps the most common understanding of the interaction between the CPE and DA in BNNs is that DA increases the effective dataset size. From Izmailov et al. [2021]: "intuitively, data augmentation increases the amount of data observed by the model, and should lead to higher posterior contraction". From Osawa et al. [2019]: "DA increases the effective sample size". From, Noci et al. [2021]: "while data augmentation may increase the amount of data seen by the model, that increase is certainly not equal to the number of times each data point is augmented (after all, augmented data is not independent from the original data)."

In this work, we seek to understand whether the commonly used, but invalid DA likelihood can cause the CPE. Our contributions are as follows.

1. We give a formal argument that the notion that DA increases the effective dataset size is flawed (Sec. 3.1).

2. We motivate the need for multi-sample bounds, by showing that previous single-sample bounds on the likelihood are equivalent to averaging log-likelihoods, which is known to be problematic (Eq. 20).

3. We derive a set of multi-sample lower bounds on the log-likelihood of a BNN incorporating DA (Sec. 3.2). These bounds are tighter than existing single-sample estimators for BNNs [e.g. Wenzel et al., 2020] and can be applied to a broad class of likelihood functions [unlike Van der Wilk et al., 2018].

4. We introduce a "finite orbit"[1] setting with a small number of admissible augmentations which allows us to compute *exact* log-likelihoods (Sec. 3.3).

5. We empirically evaluate the performance of the multi-sample bounds in both SGD training and BNN inference for image classification tasks (Sec. 4). In the latter case we explore the impact of both the bounds and the exact finite orbit likelihood on the CPE.

6. We find that the CPE persists even when using these principled DA likelihood bounds. This falsifies the hypothesis that the CPE is an artifact of loose bounds on the log-likelihood given by previous single-sample estimators.

We finish with some discussion summarising our findings and their reflection on the CPE (Sec. 6). Our conclusion in this work is that the CPE is not an artifact resulting from DA giving "randomly perturbed log-likelihood"'s [Izmailov et al., 2021].

Note that in the remainder of the paper, we will follow Izmailov et al. [2021] in regarding models with loose, single-

---

[1]We employ the term "orbit" from group theory and function invariance [Kondor, 2008], even though our augmentations do not always form groups. In this work, it refers to the support of augmentation distribution $\mathrm{P}(\mathbf{x}'|\mathbf{x})$.

sample bounds as "unprincipled" (from Izmailov et al. [2021], the "randomly perturbed log-likelihood does not have a clean interpretation as a valid likelihood function"). In contrast, we term models using our exact log-likelihoods or our multi-sample bounds as being "principled".

## 2 BACKGROUND

### 2.1 DATA AUGMENTATION

In supervised learning, we are interested in learning some unknown functional relationship from example input-output pairs $(\mathbf{x}_i, y_i), i = 1, \ldots, N$. Often, we have information about some form of invariance, i.e. the knowledge that the function does not change its output for certain transformations of the input. These might occasionally be true invariances, such as the identity of a molecule being invariant to rotations. But in most settings, these are so-called "soft" invariances or "insensitivities" [van der Wilk et al., 2018]. For instance, the class label for an image should not change due to small translations/crops of that image (but might change if we radically alter the image). The most basic form of DA takes advantage of this information by transforming, or augmenting, the inputs and copying the output value, to create additional input-output pairs which are then included in training. Often, the amount of additional "augmented data" can be unbounded, for example when allowable transformations are specified in a continuous range, e.g. rotations. This simple procedure has been very successful in improving performance in a wide variety of machine learning methods [Loosli et al., 2007, Krizhevsky et al., 2012, Bishop, 2006], and recent work has analysed the effect of data augmentation on invariances in the learned functions [Dao et al., 2019, Chen et al., 2020, Lyle et al., 2020].

### 2.2 BAYESIAN INFERENCE

Bayesian inference allows us to infer a distribution over neural network weights, which incorporates uncertainty induced by having finite data. Bayes prescribes a strict procedure for updating beliefs about unknown quantities in light of observed data. The model is specified by a prior on the weights $\mathrm{P}(\mathbf{w})$ and a log-likelihood, $\sum_{i=1}^{N} \mathcal{L}^i(y_i; \mathbf{w})$. Thus, the log-posterior is given by

$$\log \mathrm{P}\left(\mathbf{w}|\mathbf{X}, \mathbf{y}\right) = \log \mathrm{P}\left(\mathbf{w}\right) + \sum_{i=1}^{N} \mathcal{L}^i(\mathbf{w}) + \text{const}. \quad (3)$$

We define the no augmentation log-likelihood as

$$\mathcal{L}^i_{\text{noaug}}(y_i; \mathbf{w}) = \log \mathrm{P}\left(y_i|g(\mathbf{x}_i; \mathbf{w})\right). \quad (4)$$

where $g(\cdot; \mathbf{w})$ is the neural network.

# 3 METHODS

## 3.1 DOES DA INCREASE DATASET SIZE?

Many authors have claimed that DA increases the effective dataset size [Noci et al., 2021, Osawa et al., 2019, Izmailov et al., 2021]. Here we argue that this view leads to problems within the framework of probabilistic modelling. We can see this in the form of the resulting log-likelihood. For $K$ augmented inputs, $\mathbf{x}'_{i;k}$, we can write the log-likelihood for a single underlying image as,

$$\mathcal{L}^i_{\text{add}}(y_i; \mathbf{w}) = \sum_{k=1}^{K} \log \mathrm{P}\left(y_i \big| g(\mathbf{x}'_{i;k}; \mathbf{w})\right). \quad (5)$$

For continuous transformations such as rotations, there are an infinite number of possible augmentations, $K = \infty$, so $\mathcal{L}_{\text{add}}$ would result in the prior being ignored during inference. While this result seems strange, if the outputs for all augmentations $\mathbf{x}'_{i;k}$ were independently labelled (or if all the labels were correct) we would indeed have an infinitely large (conditionally) IID dataset and ignoring the prior would be the right answer. However, in practice, the unaugmented input $\mathbf{x}_i$ is labelled by an annotator who sometimes makes mistakes [Peterson et al., 2019], the result $y_i$ is assumed to apply to all augmentations $\mathbf{x}'_{i;k}$. As such, the labels for different augmentations of the same input are not independent, and an approach (such as this one) which assumes they are cannot be valid.

A method which avoids having to specify the augmented dataset size is to average the log-likelihood over the augmentation distribution $\mathrm{P}(\mathbf{x}'_i|\mathbf{x}_i)$

$$\mathcal{L}^i_{\text{loss}}(y_i; \mathbf{w}) = \mathbb{E}\left[\log \mathrm{P}\left(y_i | g(\mathbf{x}'_i; \mathbf{w})\right)\right]. \quad (6)$$

Indeed, most implementations which use DA when training BNNs target this log-likelihood, at least implicitly. They do so by taking a pre-existing inference algorithm and replacing the original input, $\mathbf{x}_i$, with a random augmentation, $\mathbf{x}'_i$. This approach is convenient, as a single sample from the augmentation distribution can provide an unbiased estimate $\hat{\mathcal{L}}_{\text{loss}} = \log \mathrm{P}\left(y_i | g(\mathbf{x}'_i; \mathbf{w})\right)$. Importantly though, a valid likelihood should arise from a valid distribution over labels, and should therefore normalize if we sum over labels. For instance, without augmentation,

$$1 = \sum_{y_i=1}^{Y} \exp\left(\mathcal{L}^i_{\text{noaug}}(y_i; \mathbf{w})\right). \quad (7)$$

However, if we try to interpret $\mathcal{L}^i_{\text{loss}}(y_i; \mathbf{w})$ as a log-likelihood we find that it does not normalize to 1,

$$1 \neq \sum_{y_i=1}^{Y} \exp\left(\mathcal{L}^i_{\text{loss}}(y_i; \mathbf{w})\right). \quad (8)$$

and therefore $\mathcal{L}^i_{\text{loss}}(y_i; \mathbf{w})$ cannot be the log of a valid probability distribution. Note that we might try to get a valid

likelihood by including a normalizer. The problem is that this normalizer would need to depend on $\mathbf{w}$, and thus would need to be included in the log-likelihood, and of course no normalizer terms appear in the loss (Eq. 6). While we could renormalize $\mathcal{L}^i_{\text{loss}}/\mathrm{LogSumExp}_{\mathcal{Y}}\left(\mathcal{L}^i_{\text{loss}}\right)$ to ensure validity, we will see in the next section that the form of $\mathcal{L}^i_{\text{loss}}$ constitutes an unnecessarily slack bound on a principled log-likelihood, which we can tighten significantly.

## 3.2 TIGHTER LOWER BOUNDS ON THE LOG-LIKELIHOOD OF PRINCIPLED DA MODELS

To incorporate DA into BNN likelihoods, we define the probabilities for each class as being averages over augmentations. We can choose to either average logits (equal to the neural network outputs, $\mathbf{f}(\cdot; \mathbf{w})$) or predictive probabilities (softmax $\mathbf{f}(\cdot; \mathbf{w})$),

$$\mathbf{p}_{\text{inv}}(\mathbf{x}_i; \mathbf{w}) = \mathbb{E}\left[\text{softmax}\,\mathbf{f}(\mathbf{x}'_i; \mathbf{w})\right], \quad (9)$$

$$\mathbf{f}_{\text{inv}}(\mathbf{x}_i; \mathbf{w}) = \mathbb{E}\left[\mathbf{f}(\mathbf{x}'_i; \mathbf{w})\right]. \quad (10)$$

where we take expectations over $\mathrm{P}(\mathbf{x}'_i|\mathbf{x}_i)$. Remember that $\mathbf{f}(\mathbf{x}'_i; \mathbf{w})$ is the (vector-valued) neural network output for an augmented input, which is used as the logits in classification, so $\mathbf{f}_{\text{inv}}(\mathbf{x}_i; \mathbf{w})$ is the outputs averaged over all augmentations of the same underlying image. Likewise, $\mathbf{p}_{\text{inv}}(\mathbf{x}_i; \mathbf{w})$ is the vector of probabilities given by averaging the predicted probabilities over augmentations. These are denoted "inv" for invariant, because averaging over augmentations can give invariances in $\mathbf{f}_{\text{inv}}(\mathbf{x}_i; \mathbf{w})$ and $\mathbf{p}_{\text{inv}}(\mathbf{x}_i; \mathbf{w})$ that are not present in the underlying neural network, $\mathbf{f}(\mathbf{x}_i; \mathbf{w})$. The resulting (usually intractable) log-likelihoods are

$$\mathcal{L}^i_{\text{prob}}(y_i; \mathbf{w}) = \log \mathrm{P}_{\text{prob}}(y_i|\mathbf{x}_i, \mathbf{w})$$
$$= \log \mathbb{E}\left[\text{softmax}_{y_i}\,\mathbf{f}(\mathbf{x}'_i; \mathbf{w})\right], \quad (11)$$

$$\mathcal{L}^i_{\text{logits}}(y_i; \mathbf{w}) = \log \mathrm{P}_{\text{logits}}(y_i|\mathbf{x}_i, \mathbf{w})$$
$$= \log \text{softmax}_{y_i}\,\mathbb{E}\left[\mathbf{f}(\mathbf{x}'_i; \mathbf{w})\right]. \quad (12)$$

These likelihoods were originally proposed in [Wenzel et al., 2020] for averaging probabilities and [van der Wilk et al., 2018] for averaging logits. However, they are intractable, as it is not (usually) possible to evaluate the expectation under all data augmentations. Instead, we need to choose an estimator or bound on these quantities. [Wenzel et al., 2020] suggested a loose single sample bound for averaging probabilities, and [van der Wilk et al., 2018] suggested an unbiased estimator that is restricted to quadratic log-likelihoods. In contrast, we show that we can get tight, intuitive and easy to evaluate, multi-sample bounds analogous to those in [Burda

et al., 2015],

$$\hat{\mathcal{L}}^i_{\text{prob},K}(y_i;\mathbf{w}) = \log\left(\tfrac{1}{K}\sum_{k=1}^{K}\text{softmax}_{y_i}\,\mathbf{f}(\mathbf{x}'_{i;k};\mathbf{w})\right),$$

$$\hat{\mathcal{L}}^i_{\text{logits},K}(y_i;\mathbf{w}) = \log\text{softmax}_{y_i}\left(\tfrac{1}{K}\sum_{k=1}^{K}\mathbf{f}(\mathbf{x}'_{i;k};\mathbf{w})\right).$$
(13)

To prove the lower bound for averaging probabilities, we first rewrite the expectation inside the logarithm of (Eq. 11) as the expectation of its average, over $K$ identically distributed random variables, $\mathbf{x}'_{i;k}$. We then take an approach familiar from variational inference [Jordan et al., 1999] by applying Jensen's inequality to the (concave) logarithm function.

$$\begin{aligned}\mathcal{L}^i_{\text{prob}}(y_i;\mathbf{w}) &= \log\mathbb{E}\left[\tfrac{1}{K}\sum_{k=1}^{K}\text{softmax}_{y_i}\,\mathbf{f}(\mathbf{x}'_{i;k};\mathbf{w})\right]\\ &\geq \mathbb{E}\left[\log\tfrac{1}{K}\sum_{k=1}^{K}\text{softmax}_{y_i}\,\mathbf{f}(\mathbf{x}'_{i;k};\mathbf{w})\right]\\ &= \mathbb{E}\left[\hat{\mathcal{L}}^i_{\text{prob},K}(y_i;\mathbf{w})\right].\end{aligned}$$
(14)

For averaging logits, we follow a similar method, noting that $\log\text{softmax}_{y_i}$ is a concave function [Boyd et al., 2004] taking a vector of logits and returning a scalar log-probability for class $y_i$. As such, we can again apply Jensen's inequality,

$$\begin{aligned}\mathcal{L}^i_{\text{logits}}(y_i;\mathbf{w}) &= \log\text{softmax}_{y_i}\,\mathbb{E}\left[\tfrac{1}{K}\sum_{k=1}^{K}\mathbf{f}(\mathbf{x}'_{i;k};\mathbf{w})\right]\\ &\geq \mathbb{E}\left[\log\text{softmax}_{y_i}\,\tfrac{1}{K}\sum_{k=1}^{K}\mathbf{f}(\mathbf{x}'_{i;k};\mathbf{w})\right]\\ &= \mathbb{E}\left[\hat{\mathcal{L}}^i_{\text{logits},K}(y_i;\mathbf{w})\right].\end{aligned}$$
(15)

Finally, note that these objectives naturally correspond to the notions of averaging logits or averaging probabilities, which could be motivated using non-probabilistic considerations. Importantly, we do not claim the notion of averaging probabilities or averaging logits for different augmentations as a contribution in itself. We only claim as a contribution the notion that averaging probabilities or logits provide lower-bounds on principled log-likelihoods including DA, implying they can be used in a principled Bayesian setting, and they are not ruled out despite having some degree of stochasticity.

Increasing $K$ reduces the variance and tightens the bounds which eventually become exact as $K \to \infty$ [Burda et al., 2015].

$$\mathbb{E}\left[\hat{\mathcal{L}}^i_{\text{logits},K}(y_i;\mathbf{w})\right] \leq \mathbb{E}\left[\hat{\mathcal{L}}^i_{\text{logits},K+1}(y_i;\mathbf{w})\right] \quad (16)$$

$$\mathcal{L}^i_{\text{logits}}(y_i;\mathbf{w}) = \lim_{K\to\infty}\hat{\mathcal{L}}^i_{\text{logits},K}(y_i;\mathbf{w}) \quad (17)$$

$$\mathbb{E}\left[\hat{\mathcal{L}}^i_{\text{prob},K}(y_i;\mathbf{w})\right] \leq \mathbb{E}\left[\hat{\mathcal{L}}^i_{\text{prob},K+1}(y_i;\mathbf{w})\right] \quad (18)$$

$$\mathcal{L}^i_{\text{prob}}(y_i;\mathbf{w}) = \lim_{K\to\infty}\hat{\mathcal{L}}^i_{\text{prob},K}(y_i;\mathbf{w}) \quad (19)$$

However, larger $K$ introduces greater computational cost. We therefore consider what value of $K$ is likely to be sensible, by plotting the bound against $K$. We indeed found that

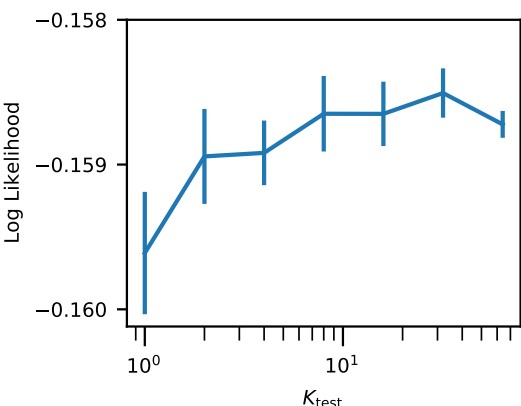

Figure 1: The effect of $K_{\text{test}}$ on the log-likelihood bound for a test batch (size 512) of CIFAR-10. Values shown for ResNet20 BNN trained and tested with $\hat{\mathcal{L}}_{\text{prob},K}$ ($K_{\text{train}} = 8$ and $T = 0.001$). Error bars cover two standard errors above/below mean for DA sampling with different seeds. Sixty seeds used for $K_{\text{test}} = \{1,2\}$, thirty for $K_{\text{test}} = 4$ and five for all other $K_{\text{test}}$.

the bound increases with $K$ up to around 10, when it saturates (Fig. 1). While these differences might seem small when evaluated purely at test-time, they seem to cause much larger differences when integrated into training (Figs. 2 and 4). In contrast, in VI, practitioners frequently use a single-sample bound. However, VI incorporates a highly effective variance reduction strategy that is absent in our setting: an optimized variational approximate posterior (see Appendix B). In principle, similar variance reduction strategies exist in our setting, but would involve learning a separate variance-reducing augmentation distribution for each image, which is clearly impractical. Indeed, in our setting, $K = 1$ represents such a crude approximation that it collapses the differences between averaging probabilities, logits, and losses,

$$\hat{\mathcal{L}}^i_{\text{logits};1}(y_i;\mathbf{w}) = \hat{\mathcal{L}}^i_{\text{prob};1}(y_i;\mathbf{w}) = \hat{\mathcal{L}}^i_{\text{loss}}(y_i;\mathbf{w}) \quad (20)$$

which are all equal to $\log\text{softmax}_{y_i}\,\mathbf{f}(\mathbf{x}'_{i;1};\mathbf{w})$.

### 3.3 FINITE ORBIT

Finally, all of the above is for the usual "full orbit" setting, where there is a distribution over a very large, or even infinite number of possible augmentations. The full orbit setting necessitates the use of the bound in (Eq. 13), and allows us to use different numbers of samples at test and training time, $K_{\text{test}}$ and $K_{\text{train}}$ respectively. Remarkably, if we consider an alternative "finite orbit" by restricting the augmentations to a small subset, we can *exactly* evaluate the log-likelihood. In the finite orbit setting, the distribution over augmented images, $\mathbf{x}'_i$, conditioned on the underlying unaugmented

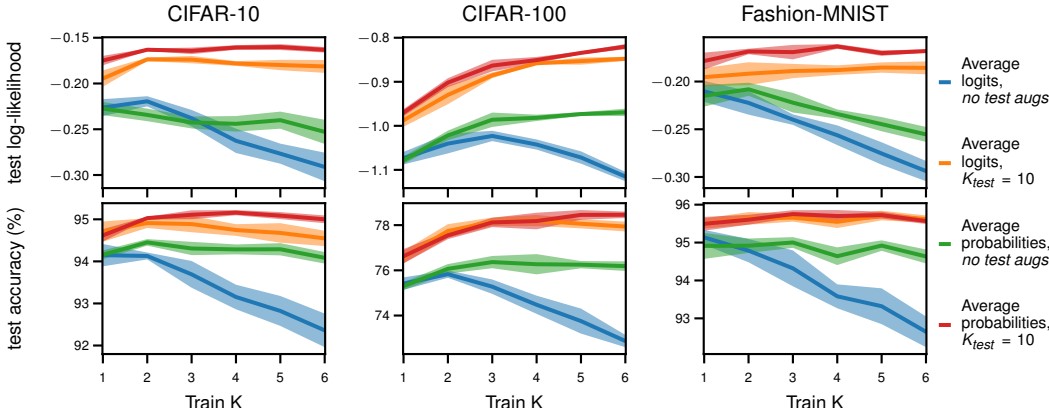

Figure 2: Comparison of averaging logits and probabilities for different values of $K_{\text{train}}$, and using $K_{\text{test}} = 10$ vs. using no test-time augmentations. Here, we use ResNet18 with SGD (i.e. no Bayesian inference). We use only full orbit to decouple $K_{\text{train}}$ from $K_{\text{test}}$.

image, $\mathbf{x}_i$, can be written as,

$$P\left(\mathbf{x}_i'|\mathbf{x}_i\right) = \frac{1}{K}\sum_{k=1}^{K}\delta\left(\mathbf{x}_i' - a_k(\mathbf{x}_i)\right), \quad (21)$$

where $\delta$ is the Dirac-delta, and $a_k$ is a function that applies the $k$th fixed augmentation. In this setting, it is possible to exactly compute $\mathcal{L}_{\text{logits}}^i(y_i; \mathbf{w})$ and $\mathcal{L}_{\text{prob}}^i(y_i; \mathbf{w})$ by summing over the $K$ augmentations. This allows us to empirically explore how exact log-likelihood computation influences the CPE, comparing it with the bounds in the full-orbit setting (Eq. 13) in Sec. 4.2. When implementing finite orbit augmentation in practice, we choose the $K$ fixed augmentations by sampling them before training. The finite orbit setting uses the same augmentations, and therefore the same number of augmentations, at test and train time: $K_{\text{train}} = K_{\text{test}} = K$.

## 4 RESULTS

### 4.1 PRINCIPLED DA IN NON-BAYESIAN NETWORKS

We begin by comparing averaging logits and averaging probabilities in a non-Bayesian setting: SGD. Critically, higher values of $K_{\text{train}}$ imply a larger computational cost per epoch, as each image is replicated and augmented $K_{\text{train}}$ times before going through the network. When assessing the benefit of averaging probabilities/logits over standard DA for SGD training, we must therefore control for computational cost. We do this by training for $200/K_{\text{train}}$ epochs. Note that $K_{\text{train}} = 1$ with no test-time augmentation (i.e. green and blue in Fig. 2) corresponds to the standard DA approach for both averaging logits and averaging probabilities (Eq. 20). In this experiment, we consider only full orbit, which unlike finite orbit allows us to decouple $K_{\text{train}}$ and $K_{\text{test}}$.

We trained ResNet18[2] on CIFAR-10, CIFAR-100 [Krizhevsky et al., 2009][3] and FashionMNIST [Xiao et al., 2017][4] with a learning rate of 0.1, decayed to 0.01 three quarters of the way through training. We apply the same two augmentation transformations as Wenzel et al. [2020], Fortuin et al. [2021b], Noci et al. [2021]: 1. a random crop with padding of four pixels on all borders and 2. a random horizontal flip with probability 0.5. The training runs took around 12 GPU-days on Nvidia 2080s.[5]

In agreement with past work [Lyle et al., 2020], we found that averaging over augmentations at test-time (red and orange) is better than using the test image without augmentation (green and blue), with $K_{\text{train}} = 1$ corresponding to the standard DA procedure. In addition, we show that improved performance with multiple test-time augmentations continues to hold for larger values of $K_{\text{train}}$. Thus, if sufficient compute is available at test-time, averaging across augmentations gives an easy method to improve the performance of a pre-trained network.

Importantly, we see some performance gains with higher values of $K_{\text{train}}$ if we focus on the case with test augmentations, though they are somewhat inconsistent across datasets. We see strong improvements for the hardest dataset (CIFAR-100), and smaller improvements that saturate at $K_{\text{train}} = 2$ for CIFAR-10. For FashionMNIST, the picture is more mixed. We suspect this is because we used a DA strategy tuned for CIFAR-10 and CIFAR-100, rather than FashionMNIST.

---

[2]github.com/kuangliu/pytorch-cifar; MIT Licensed

[3]cs.toronto.edu/~kriz/cifar.html

[4]github.com/zalandoresearch/fashion-mnist; MIT Licensed

[5]Code available: anonymous.4open.science/r/Augmentations-1E35/

In addition, averaging probabilities seems to give somewhat better performance than averaging logits: compare averaging probabilities vs. logits both with test-time augmentation (red vs. orange) and without test-time augmentation (green vs. blue). The performance differences are consistent in both comparisons, though smaller when test-time augmentation is applied.

Indeed, performance falls quite dramatically as $K_{\text{train}}$ increases for averaging logits without test-time augmentation (blue). This is an indication that averaging probabilities and logits might actually behave quite differently. To understand how these differences might arise, consider the effect of averaging on the NN function itself. Both schemes can be justified by using averaging to increase invariance to the augmentation transformations (Sec. 3.2). Averaging probabilities, however, also forces the NN function itself to become invariant. If different augmentations produce different predictions, then the resulting averaged class probabilities will be more uncertain, which is penalized by the likelihood on the training points. This effect is much weaker when averaging logits. Consider an extreme example, as illustrated in Fig. 3. It is a two-class classification problem with two augmentations, $\mathbf{x}_1'$ and $\mathbf{x}_2'$, of the same image with logits, $\mathbf{f}(\mathbf{x}_1') = (10, -10)$ and $\mathbf{f}(\mathbf{x}_2') = (-1, 1)$. Averaging logits gives us $\mathbb{E}\left[\mathbf{f}(\mathbf{x}')\right] = (4.5, -4.5)$, and applying the softmax, we very confidently predict the first class. In contrast, if we use averaging probabilities, then the first augmentation almost certainly predicts the first class $p(\mathbf{x}_1') \approx (1, 0)$ and the second augmentation almost certainly predicts the second class, $p(\mathbf{x}_2') \approx (0, 1)$, so when we average them we obtain $\mathbb{E}\left[p(\mathbf{x}')\right] \approx (0.5, 0.5)$, which indicates a high degree of uncertainty.

## 4.2 BAYESIAN NEURAL NETWORKS AND THE COLD POSTERIOR EFFECT

Next, we ask a very different question: how is the CPE influenced when DA is incorporated into the model in a principled way? To this end, we use a different experimental setup. In particular, we take the code[6] and networks from Fortuin et al. [2021b,a] and mirror their experimental setup for CIFAR-10 and MNIST as closely as possible. This code combines a cyclical learning rate schedule [Zhang et al., 2019], a gradient-guided Monte Carlo (GGMC) scheme [Garriga-Alonso and Fortuin, 2021], and the preconditioning and convergence diagnostics from Wenzel et al. [2020]. The CIFAR-10 DA transformations are the same as those described in Sec. 4.1 and for MNIST we apply random cropping with a padding of two pixels, then random rotation by an angle sampled uniformly over $(-\pi/6, \pi/6)$. Following Fortuin et al. [2021b], we ran 60 cycles with 50 epochs in each cycle. We recorded one sample at the end of each of

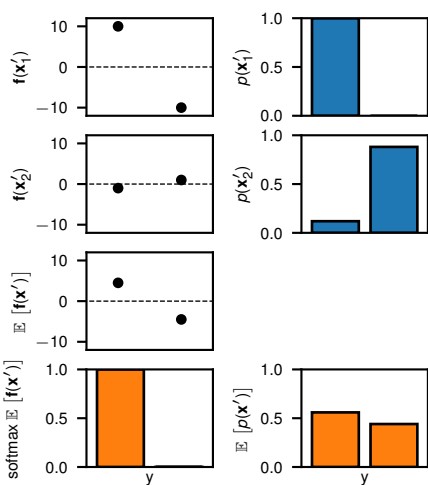

Figure 3: Example effect of averaging logits against averaging probabilities. $\mathbf{x}_1'$ and $\mathbf{x}_2'$ are two augmentations of the same image, $\mathbf{f}(\mathbf{x}_1')$ and $\mathbf{f}(\mathbf{x}_2')$ are logits outputted by a NN, and $p(\mathbf{x}_1')$ and $p(\mathbf{x}_2')$ are the probabilities corresponding to these logits. The prediction derived from the averaged logits is much more certain than the average of the individual probabilities.

the last five epochs of a cycle, giving 300 samples total.

Importantly, to allow for running many sampling epochs in these experiments, we follow Fortuin et al. [2021b] in using the ResNet20 architecture from Wenzel et al. [2020] for CIFAR-10, which has far fewer channels than the ResNet18 used in Sec. 4.1 (i.e. 32 channels for the first block up to 128 in the last block compared to 64 channels up to 512 [He et al., 2016a]). As such, SGD with this network performs poorly compared with that in Sec. 4.1 (ResNet20, CIFAR-10 accuracy $\sim 92\%$ [Wenzel et al., 2020] vs. ResNet18, CIFAR-10 accuracy $\sim 95\%$ [He et al., 2016b]). For MNIST, we use the three-layer fully connected network (FCNN) used by Fortuin et al. [2021b]. The experiments took around 90 GPU-days on Nvidia RTX6000s[7].

The results are presented in Fig. 4. We replicate the finding that the CPE is largely absent without DA (dashed black line), and is present in the standard setup with DA at training time ($K_{\text{train}} = 1$) but without augmentation at test time (solid black). Further, we show that the CPE persists with principled DA likelihoods: averaging logits with full orbit (purple, first and third rows), and averaging probabilities with finite and full orbits (green).

For CIFAR-10, the best method overall appears to be averaging probabilities with a full orbit (dark green line, third row) at $T = 0.001$, though at $T = 1$ averaging logits (dark purple lines) outperforms the other methods. For the MNIST experiments, logit averaging over a full orbit (purple line,

---
[6]github.com/ratschlab/bnn_priors; MIT Licensed

---
[7]Code available: https://github.com/sethnabarro/bnn-data-aug/

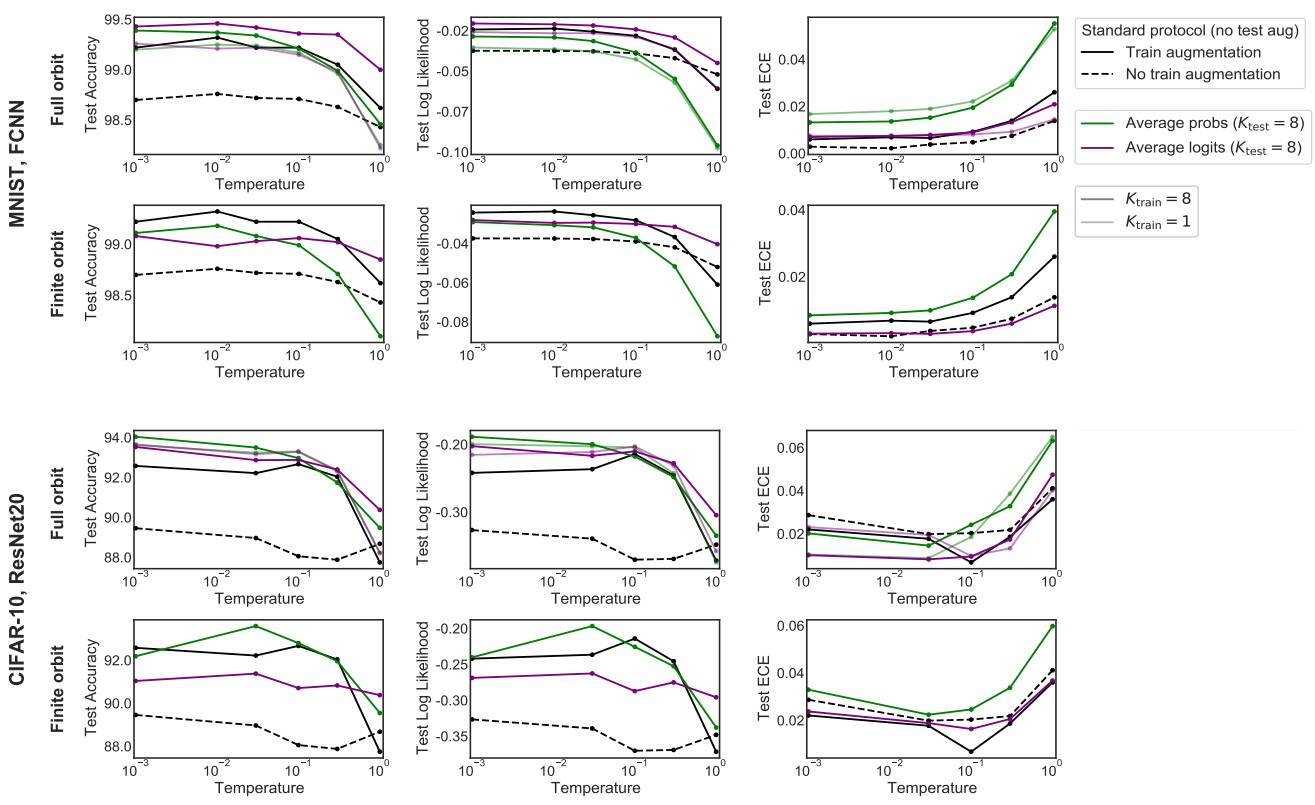

Figure 4: The cold posterior effect for different DA setups with GGMC inference [Garriga-Alonso and Fortuin, 2021]. Without DA, there is a minimal CPE. Most other configurations show significant improvement for $T < 1$, with the exception of averaging the logits over a finite orbit. Averages computed with $K_{\text{train}} = 8$ and $K_{\text{test}} = 8$.

top row) performs best at all temperatures, though has a similar accuracy to averaging probabilities (green line, top row) at $T = 0.001$. Interestingly, the CPE for averaging probabilities (green) is stronger than that for both logit averaging (dark purple) and standard DA (solid black), across all MNIST experiments.

For both datasets, the CPE is near absent in one particular setting: averaging logits with a finite orbit (purple line, second and fourth rows). However, the relevance of this is unclear, as for CIFAR-10 it is clearly the worst performing of all DA approaches, and for MNIST it is outperformed by standard DA. Indeed, remember that the arguments for the optimality of Bayesian inference apply only in the case that the model is well-specified [Kolmogorov, 1950, Savage, 1954, Jaynes, 2003]. However, the comparatively poor performance of averaging logits with a finite orbit indicates that it is likely to be the wrong model, while other settings are likely to be closer to the true model. In that case, the presence or absence of the CPE in the wrong model (averaging logits with a finite orbit) is immaterial to our understanding of the CPE in the right model. Note that this argument could not be made if there was a model without the CPE with performance equal to or better than the other models (see

Sec. 6 for further discussion).

The CPE was originally discovered in Wenzel et al. [2020] when assessing test accuracy and log-likelihood — they did not consider other measures of distribution calibration like expected calibration error (ECE). Indeed, later work on the CPE found that measures such as ECE are far more complex and usually do not agree with test accuracy and log-likelihood [Fortuin et al., 2021a]. It is therefore difficult to interpret the differences between test log-likelihod and ECE, especially if we remember that test log-likelihood is itself a proper scoring rule [Gneiting and Raftery, 2007], and therefore captures one possible notion of calibration. In particular, test log-likelihood heavily penalizes an event assessed as low probability actually happening, e.g. if our classifier predicts a probability of $0.001\%$, while the actually happens even $0.1\%$ of the time. In contrast, ECE considers the absolute difference in probability, so it far more heavily penalises e.g. a predicted probability of $40\%$ while the event actually happens $60\%$ of the time. Needless to say, the most appropriate measure of calibration will depend heavily on the domain, with log-likelihood being more appropriate for low-probability but high risk events. In our CIFAR-10 experiments, averaging probabilities (green) achieves the

greatest log-likelihood scores, standard DA (solid black) achieves the lowest ECE. This is contrasted with MNIST, for which averaging logits (purple) has highest log-likelihood and no DA (dashed black) has lowest ECE.

The usefulness of our results is contingent on understanding whether we are indeed accurately approximating the posterior. To check this, we computed the kinetic temperature [Leimkuhler and Matthews, 2015], which estimates the temperature of a given parameter in the Langevin dynamics simulation from the norm of its momentum. In expectation, the kinetic temperature estimator should be equal to the desired temperature, $T$. The results (Appendix C) indicate that all the samplers run at their desired temperature, a result that is consistent with accurate posterior sampling.

As discussed in Sec. 3, increasing $K$ tightens our log-likelihood bounds, but incurs greater computational cost. It is natural to question which value of $K$ is a good trade-off. We explore how the log-likelihood of test data under a trained model varies with $K_{\text{test}}$. As expected, the results (Fig. 1) show the log-likelihood increases with $K_{\text{test}}$, with even $K = 2$ being a significant improvement over $K = 1$ (standard DA). However, the curve plateaus, suggesting that for CIFAR-10, there is little benefit of using $K > 8$.

## 5  RELATED WORK

Past work introduced noisy-input generative models which average probabilities [Wenzel et al., 2020]. However, this work did not consider the tighter multi-sample bounds developed here, or the finite orbit setting which allows us to evaluate the exact likelihood. This left open the possibility raised by Izmailov et al. [2021] that the CPE was an artifact of standard DA resulting in an invalid likelihood. In contrast, we considered exact likelihoods in the finite orbit setting, and tighter multi-sample lower bounds in the full orbit setting. Further, the invariant function perspective allowed us to derive a log-likelihood bound for averaging logits, not considered by Wenzel et al. [2020]. As the CPE persists when using our principled DA models, we can exclude the possibility that the CPE is an artifact of DA giving a "randomly perturbed log-likelihood". Other work has introduced a log-likelihood estimator for averaging GP logits using the invariance principle [van der Wilk et al., 2018]. However, the method only works for a quadratic log-likelihood and thus necessitates Pólya-Gamma approximations for classification. Further, the work did not consider BNNs or the connection to the CPE.

There is a small but growing body of work that considers averaging over multiple augmentations at training time [Hoffer et al., 2019, Berman et al., 2019, Choi et al., 2019, Benton et al., 2020, Lyle et al., 2020, Touvron et al., 2021, Fort et al., 2021]. However, this work was not done within a Bayesian framework (e.g. by using stochastic gradient Langevin dynamics (SGLD) or a similar inference algorithm), did not show that averaging across multiple training augmentations gives a multi-sample bound on the log-likelihood of a principled model, did not consider the finite-orbit setting where the log-likelihood can be computed exactly, and did not consider the interaction with the CPE. In addition, much of this work uses averaging losses [Hoffer et al., 2019, Berman et al., 2019, Choi et al., 2019, Benton et al., 2020, Touvron et al., 2021, Fort et al., 2021] which is equivalent to using a loose single-sample bound on the log-likelihoods. While Lyle et al. [2020] show that feature averaging during training can improve generalization, our work is, to the best of our knowledge, the first to average predicted probabilities at training time. Finally, the idea of averaging at test-time is more common and has been practiced for longer [e.g. Krizhevsky et al., 2012, Simonyan and Zisserman, 2014, He et al., 2015, Szegedy et al., 2015, Foster et al., 2020].

A considerable body of past work on BNNs uses DA, both with variational inference [Blundell et al., 2015, Zhang et al., 2018, Osawa et al., 2019, Ober and Aitchison, 2020, Unlu and Aitchison, 2021], Laplace approximations [Immer et al., 2021] and SGLD [e.g. Zhang et al., 2019, Fortuin et al., 2021b, Wang and Aitchison, 2021]. However, as discussed in Sec. 2 (Background), these methods simply substitute non-augmented for augmented data and thus do not use a valid log-likelihood. In contrast, we incorporated DA into the probabilistic generative model, and thus are able to give valid log-likelihoods based on averaging logits or averaging probabilities in the classification case.

## 6  CONCLUSION

We have shown how DA can be properly incorporated into a generative model suitable for BNN inference, by deriving a lower-bound on the log-likelihood of the augmentation-averaged network output. Empirically, we have seen that the CPE persists even when using our principled DA formulation, and in agreement with past work [Wenzel et al., 2020, Fortuin et al., 2021b, Izmailov et al., 2021], we show that the CPE disappears without DA.

What do these results imply for the origin of the CPE? Our models in principle have a clean log-likelihood which can be evaluated exactly in the finite orbit setting, or which we estimate using tightened multi-sample bounds in the full orbit setting. This falsifies the hypothesis, that the CPE is an artifact arising from DA giving a "randomly perturbed log-likelihood [which] does not have a clean interpretation as a valid likelihood function".

Indeed, it is worth stepping back and considering the original motivation for studying the CPE, namely that if we have the correct model, then Bayesian inference with $T = 1$ should give optimal performance [Kolmogorov, 1950, Savage, 1954, Jaynes, 2003, Wenzel et al., 2020]. Critically, we

need the right model for us to expect optimal performance at $T = 1$. We now have two classes of model, with DA and without DA, so which is right(er)? Given the significant and widely recognised performance benefits of DA, it seems very likely that the "right" model would include some form of DA. If the model with DA is right(er), and that model displays the CPE, then the CPE still demands an explanation, and the presence or absence of the CPE in the wrong model without DA is immaterial. As such, the presence of the CPE in models with DA remains an important problem, and is likely to be caused by one of the two other explanations discussed in Sec. 1 (Introduction): either data curation [Aitchison, 2020] or prior misspecification [Wenzel et al., 2020, Fortuin et al., 2021b]. Indeed, we would tentatively suggest the opposite of Izmailov et al. [2021]: that it is in reality the *lack* of a CPE without DA that is an artifact of using the wrong model (i.e. without DA).

Finally, note that the CPE is not always observed, e.g. in language classification [Izmailov et al., 2021]. This is absolutely expected as the data-curation explanation of Aitchison [2020] only implies CPE in fairly restricted settings; i.e. *only* in the case of reasonably accurate approximate posterior inference, such as SGLD, in a BNN where the data has been curated by excluding datapoints with an ambiguous class-label. Thus, Aitchison [2020] does not lead us to expect the CPE e.g. in latent variable models, in regression settings (where you typically do not curate data), or in hybrid models where we perform Bayesian inference over only a small subset of parameters.

## Acknowledgements

VF was supported by the Swiss Data Science Center, the Swiss National Science Foundation, and St. John's College Cambridge.

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
