# OpenReview forum: "Data augmentation in Bayesian neural networks and the cold posterior effect"
_auai.org/UAI/2022/Conference — UAI 2022 Poster_

### Official Review · Reviewer_tbyQ · 2022-03-29

**Q2(1) Originality/Novelty:** 2
**Q2(2) Significance/Impact:** 2
**Q2(3) Correctness/Technical Quality:** 3
**Q2(6) Clarity Of Writing:** 3
**Q6 Overall Score:** 4
**Q8 Confidence In Your Score:** 4

**Q1 Summary And Contributions:**

This paper provides an analysis of principled data augmentation formulations for Bayesian neural networks in the context of cold posterior effect. The authors argued that taking the expectation of the log-likelihood loss w.r.t. augmented data is not principled. Instead they propose to take the expecation of either the probabilities or the logits which admit tighter lower bound using multiple DA samples. The authors observe that the cold posterior effect still persists in these cases.

**Q2 Assessment Of The Paper:**

More detailed information regarding each of these aspects is given below:

**Q2(4) Quality Of Experiments (Optional):**

2: Fair: The experimental evaluation is weak: important baselines are missing, or the results do not adequately support the main claims.

**Q2(5) Reproducibility:**

3: Good: Key resources (e.g., proofs, code, data) are available and key details (e.g., proofs, experimental setup) are sufficiently well-described for competent researchers to confidently reproduce the main results.

**Q3 Main Strengths:**

- The paper is easy to follow.
- Source code with detailed instruction is provided.

**Q4 Main Weakness:**

- the novelty of the paper is lacking
- the paper only provides limited insights w.r.t. the cold posterior effect

**Q5 Detailed Comments To The Authors:**

A main issue I see in this paper is the novelty, since this paper brings little contribution beyond existing works. In the methodology section (section 3), section 3.1 is mainly a developed discussion that reaches the same conclusion as prior work, “randomly perturbed log-likelihood does not have a clean interpretation as a valid likelihood function”, mentioned in section 1 of the paper. The authors then propose the likelihood formulations with averaging probabilities/logits, both already existing works, then show that they admits tight  multi-sample lower bounds, which is again already known from the "Importance weighted autoencoders" paper by Burda et al.

Secondly, this work offers little insights in to the cold posterior effect beyond the observation that 2 existing likelihood formulation that incorporate data augmentation through averaging probabilities/logits does not prevent the CPE.

One minor detail, if I am not mistaken, the approaches of Zhang et al. 2018 and Osawa et al. 2019 cited in Section 5 are not related to SGLD but are rather VI based.


**Q7 Justification For Your Score:**

This paper has insufficient novelty w.r.t. existing Bayesian formulation of data augmentation. Also, the paper only provides limited insights w.r.t. the cold posterior effect.

**Q9 Complying With Reviewing Instructions:**

1: Yes.

---

### Official Review · Reviewer_xKMD · 2022-04-07

**Q2(1) Originality/Novelty:** 2
**Q2(2) Significance/Impact:** 2
**Q2(3) Correctness/Technical Quality:** 3
**Q2(6) Clarity Of Writing:** 4
**Q6 Overall Score:** 6
**Q8 Confidence In Your Score:** 3

**Q1 Summary And Contributions:**

The paper studies the cold posterior effect and its relation to data augmentation. The paper finds that the cold posterior effect is still present even when data augmentation is implemented in a principled manner and as a result argues that the cold posterior effect cannot be  attributed to data augmentation.

**Q2 Assessment Of The Paper:**

More detailed information regarding each of these aspects is given below:

**Q2(4) Quality Of Experiments (Optional):**

3: Good: The experimental evaluation is adequate, and the results convincingly support the main claims.

**Q2(5) Reproducibility:**

3: Good: Key resources (e.g., proofs, code, data) are available and key details (e.g., proofs, experimental setup) are sufficiently well-described for competent researchers to confidently reproduce the main results.

**Q3 Main Strengths:**

The main strengths are as follows:
* The clarity of the argument put forward and how the empirical results support the paper's argument.
* The maths was clear.

**Q4 Main Weakness:**

* While the experimental results support the argument made in the paper, only using CIFAR-10, CIFAR-100, and Fashion MNIST with a single ResNet20 model does reduce the strength of the argument. It makes one wonder whether these results are specific to ResNet20 models or whether they generalise to other architectures. However, I do appreciate that the commonly used Bayesian inference approaches makes larger models less viable. Maybe a similar experiment with a simple dense NN would help with this respect.
* In addition to the single architecture and three data sets being used, the paper only uses a single inference approach of Cyclic GGMC. In Izmailov's work on "What are Bayesian neural network posteriors really like?" they focus on HMC. They argue that a less approximate (although computational costly) Bayesian inference approach of HMC gives near optimal performance and therefore they don't see the CPE. It would have been nice to see other inference approaches used.

**Q5 Detailed Comments To The Authors:**

* I think the main concerns are that of using a single inference scheme and single architecture and making generalisations based on these results. The paper still reads well, so I think a few comments from the authors on whether they think the current experiments are sufficient.
* My understanding is that the premise of the paper is that the CPE is not caused by DA, which goes against what was noted in Izmailov et al. 2021. The results seem to show this is the case and it would be good to hear from the authors a bit more about what this means for practitioners. E.g. If I were to train a model, should I use a principled DA formulation? Will that get me better results? Also further highlighting my comment from before as to whether this applies to other Bayesian inference approaches.

**Q7 Justification For Your Score:**

* Overall the paper reads well and the experiments support the argument.
* The main reason for the borderline accept is because the paper only explores one Bayesian inference approach and one architecture. However given it reads well and was interesting I wanted to give the benefit of the doubt.

#### After rebuttal I have raised my score

**Q9 Complying With Reviewing Instructions:**

1: Yes.

---

### Official Review · Reviewer_ziEo · 2022-04-12

**Q2(1) Originality/Novelty:** 2
**Q2(2) Significance/Impact:** 2
**Q2(3) Correctness/Technical Quality:** 3
**Q2(6) Clarity Of Writing:** 4
**Q6 Overall Score:** 7
**Q8 Confidence In Your Score:** 3

**Q1 Summary And Contributions:**

The paper present a Bayesian analysis of data augmentation. Starting from the observation that usual data augmentation strategies lead to invalid log-likelihood functions, the authors propose two approaches for valid likelihoods with data augmentation.
The authors then claim that such novel approaches help understanding the cold posterior effect and suggest various insights on Bayesian neural network behavior..
The methods are evaluated empirically on standard datasets.

**Q2 Assessment Of The Paper:**

More detailed information regarding each of these aspects is given below:

**Q2(4) Quality Of Experiments (Optional):**

3: Good: The experimental evaluation is adequate, and the results convincingly support the main claims.

**Q2(5) Reproducibility:**

4: Excellent: Key resources (e.g., proofs, code, data) are available and key details (e.g., proof sketches, experimental setup) are comprehensively described for competent researchers to confidently and easily reproduce the main results.

**Q3 Main Strengths:**

- This is a well-written paper on an interesting subject that presents insights on the behavior of Bayesian neural network training and predictions.

-  appropriate experiments

- The code to reproduce the experiments is provided in a well documented anonymized repository.



**Q4 Main Weakness:**

- I am unsure on the novelty of the results. It seems that the likelihoods in Eqs (11) and (12) were already proposed in two previous works. Moreover the bounds and their results as $K \to \infty$ follows from Burda et al. (2015).

- while the subject is interesting, the conclusions and the possible impact of the paper are not clear to me: do the authors argue for the use of the proposed approach in Bayesian neural networks? or the lower bounds and finite orbit approach are only a tool to evaluate and study the cold posterior effect?

**Q5 Detailed Comments To The Authors:**

one additional comment is the wording *right model* or *wrong model* used in the paper, I find it a bit strange and I do not understand what the authors mean by that. I would like the authors to clarify this point especially.


and I have some minor questions and suggestion.

- how is precisely that with the finite orbit approach is possible to compute exact likelihood?

- improve readability of the figures: In FIgure 4 the legend and labels are small (I think that legend position could be changes and font size increased);

- I think that $K_train$ and $K_test$  are not defined, I suggest to add a sentence to define these parameters in section 3.2 (or before)

- I got a bit lost with all the possible methods and settings in sec. 4.1 and 4.2, maybe try to highlight the important results (also in the figure) and remove redundant or additional cases from the main paper.

**Q7 Justification For Your Score:**

I think the work is a solid contribution to Bayesian neural networks and data augmentation literature.
The paper is well written and clear, I think the authors make all the efforts to explain the results and the ideas behind.

**Q9 Complying With Reviewing Instructions:**

1: Yes.

---

### Official Review · Reviewer_jzmh · 2022-04-14

**Q2(1) Originality/Novelty:** 2
**Q2(2) Significance/Impact:** 2
**Q2(3) Correctness/Technical Quality:** 2
**Q2(6) Clarity Of Writing:** 1
**Q6 Overall Score:** 5
**Q8 Confidence In Your Score:** 4

**Q1 Summary And Contributions:**

The authors proposed a method to account for data augmentation (DA) from a Bayesian perspective. They also proposed tighter multi-sample bounds for the log-likelihood of a BNN incorporating DA and analyzed the existence of the cold posterior effect when using these bounds. An approximate version for the above multi-sample bounds was also provided.

**Q2 Assessment Of The Paper:**

More detailed information regarding each of these aspects is given below:

**Q2(4) Quality Of Experiments (Optional):**

3: Good: The experimental evaluation is adequate, and the results convincingly support the main claims.

**Q2(5) Reproducibility:**

3: Good: Key resources (e.g., proofs, code, data) are available and key details (e.g., proofs, experimental setup) are sufficiently well-described for competent researchers to confidently reproduce the main results.

**Q3 Main Strengths:**

This paper studies the open question of accounting data augmentation from a Bayesian perspective.

**Q4 Main Weakness:**

1.	In eq.8, when the new loss is estimated using a sample from augmentation distribution instead of the expectation variant. It looks like it is equivalent to eq.7.
2.	Although it is rightly pointed out that labels for different augmentations are not independent, the argument that data augmentation does not increase the size of the effective dataset is not conclusive. As [Wenzel et al.,2020] proposed Jensen Prior interpretation for DA, does this paper propose interpreting DA using a modified likelihood function that is an expectation over all augmentations?
3.	In eq.14, eq.15, what is the distribution over which expectation is taken?
4.	In eq.16, eq.18, the likelihoods on the left-hand and right-hand sides should be of the same type.

**Q5 Detailed Comments To The Authors:**

The paper lacks a coherent writing style in describing a section or a method. With many citations in between the sentences, it remains hard to read the paper.


**Q7 Justification For Your Score:**

The score mainly stems from the weaknesses listed above. The main contributions of this paper are tighter multi-sample bounds and the analysis of CPE when these bounds are used. This is an incremental work of [Wenzel et al.,2020] with minor observations.


**Q9 Complying With Reviewing Instructions:**

1: Yes.

---

### Decision · Program_Chairs · 2022-05-15

**Decision:**

Accept (Poster)

**Comment:**

Meta Review: The paper provides "principled" bounds of the loglikelihood under data augmentation (DA) and argues that the cold posterior effect (CPE) persists even when using these bounds. Reviewers are overall quite positive, but do raise concerns about the novelty and whether the results warrant the quite strong conclusion. The authors are encouraged to incorporate the suggestions of the reviewers to improve their paper and in particular to clearly separate speculative conjectures (e.g., that principled DA cannot prevent CPE) from solid conclusions (e.g., that the specific objectives that follow from the multi-sample bounds still cannot prevent CPE).